# A Review of Advanced Methods for the Quantitative Analysis of Single Component Oil in Edible Oil Blends

**DOI:** 10.3390/foods11162436

**Published:** 2022-08-13

**Authors:** Xihui Bian, Yao Wang, Shuaishuai Wang, Joel B. Johnson, Hao Sun, Yugao Guo, Xiaoyao Tan

**Affiliations:** 1School of Chemical Engineering and Technology, Tiangong University, Tianjin 300387, China; 2Shandong Provincial Key Laboratory of Olefin Catalysis and Polymerization, Shandong Chambroad Holding Group Co., Ltd., Binzhou 256500, China; 3School of Health, Medical & Applied Sciences, Central Queensland University, Bruce Hwy, North Rockhampton, QLD 4701, Australia

**Keywords:** edible oil blends, sample design, instruments, chemometrics, quantitative analysis

## Abstract

Edible oil blends are composed of two or more edible oils in varying proportions, which can ensure nutritional balance compared to oils comprising a single component oil. In view of their economical and nutritional benefits, quantitative analysis of the component oils in edible oil blends is necessary to ensure the rights and interests of consumers and maintain fairness in the edible oil market. Chemometrics combined with modern analytical instruments has become a main analytical technology for the quantitative analysis of edible oil blends. This review summarizes the different oil blend design methods, instrumental techniques and chemometric methods for conducting single component oil quantification in edible oil blends. The aim is to classify and compare the existing analytical techniques to highlight suitable and promising determination methods in this field.

## 1. Introduction

Edible oil is an important ingredient used for cooking and flavoring a wide range of foods. Chemically speaking, edible oil not only provides nutrients, such as unsaturated fatty acids and fat-soluble vitamins, but also offers some essential nutritional requirements that cannot be endogenously produced by humans [1,2]. However, a single type of edible oil cannot meet the requirements of functional and nutritional balance. Therefore, it is common to mix two or more types of edible oils in different proportions to prepare edible oil blends, which can effectively overcome the nutritional shortcomings of the single oil types. Furthermore, the various fatty acids found in edible oil blends can provide protection against some severe chronic illnesses, such as neurodegenerative, inflammation and cardiovascular diseases [3]. Consequently, the practice of blending edible oils is becoming increasingly common, with sales volumes of edible oil blends continuing to rise in recent years.

Due to the different geographical sources, yields and nutritional values, the prices of different edible oils can vary widely [4]. Conventional edible oils, such as soybean, sunflower, olive, corn, and rapeseed oils, are most commonly utilized as the base oils for edible oil blends as they are relatively cheap and readily available. On the other hand, more unconventional edible oils, such as flaxseed, camellia and peony seed oils, have higher nutritional value and unique flavors, thus are more expensive than conventional edible oils. As a result of these large price variances between edible oils, some businesses market their product with strong emphasis on the presence of the high-grade oil (which is only present in small amounts) and omit the proportion of lower-grade oil [5], which can be seen as misleading consumers. Due to the similar appearance of different oils and the homogeneity of oil blends, the content of each oil cannot be distinguished visually. Therefore, in order to protect the rights and interests of consumers and maintain a fair edible oil market, it is necessary to develop reliable quantitative detection methods for the quantitative authentication of edible oil blends.

Spectroscopic-based techniques have the advantages of non-destructive testing, simple operation, rapid analysis and high sensitivity [2], and have been widely used for quantifying single component oil in edible oil blends. However, the spectra from different edible oils can be difficult to distinguish due to the similarity in their components, making quantitative analysis difficult to implement. Pairing spectroscopic techniques with chemometrics can help solve this problem. The spectral matrix is processed by chemometric methods, and multivariate calibration models are established between the spectral matrix and target values to determine the content of the single component oil. A range of multivariate calibration methods, including multiple linear regression (MLR), principal component regression (PCR), partial least square (PLS) regression and support vector regression (SVR), may be used [6]. As the spectra matrix for model establishment usually contain some useless information and hundreds of variables, preprocessing and variable selection methods are used to improve model accuracy and robustness.

There have been several recent reviews covering the quality analysis of edible oils. Salah et al. [2] reviewed different adulteration detection techniques for edible oils. Mahesar et al. [7] presented an overview of infrared spectroscopy in combination with chemometric techniques to determine the functional compounds in olive oil. Another review by Ou et al. [8] summarized the advanced sensor methods used to detect adulteration of olive oil, based on the different physical and chemical properties. Rohman et al. [9] reviewed the application of infrared spectroscopy combined with chemometrics for the authentication of various fats and oils. Thus, although specific spectroscopy techniques used for quality detection of specific oil have been summarized by previous workers, no review focuses on the quantitative analysis of single component oil in edible oil blends.

The aim of this paper was to review the existing analytical techniques for single component oil quantification in edible oil blends, and to classify and compare these techniques to find suitable and promising determination methods in this field. This review includes: (1) investigating the main sample design methods used to produce oil blends; (2) summarizing different instrumental techniques for measuring edible oil blends; (3) summarizing the commonly used preprocessing, variable selection and multivariate calibration methods used by current researchers.

## 2. Sample Design Methods

To establish and validate quantitative models for the analysis of single component oil in edible oil blends, a certain number of oil blend samples must be prepared. If the edible oil blends are directly purchased from a supermarket, it is difficult to determine the proportion of each component oil. Therefore, in order to obtain oil blends with accurately known compositions, almost all researchers use a sample design to obtain oil blend samples by mixing together designated volumes or weights of pure oils. The general process for single component oil quantification in edible oil blends is shown in Figure 1. First, the purchased pure oils are used to prepare oil blend samples with different percentages of each component oil, before the mixtures are characterized using various analytical instruments. Following this, the obtained dataset is processed by chemometric methods to create a quantitative model to predict the content of the component oil. In the quantitative analysis of single component oil in edible oil blends, the difference among different analytical techniques is the selection of instruments for signal measurement, while the oil blend sample preparation for model establishment, chemometric methods and implementation software are the same. Therefore, the cost of different analytical techniques varied with the price of the different instruments.

In this process, the first step is to design a sufficient number of oil blend samples. Designing the proportions of each oil is more difficult compared to other situations, such as solutions comprising a solvent containing different solute concentrations. However, for oil blend samples, the sum of the oil contents in each sample must be 100%. If the content of each component oil is designed to range between 0–100% and the restriction of the total oil content is not considered, the total oil content could exceed 100%. Consequently, a logical approach is required to design the content of each oil in oil blend samples. At present, the most common methods of preparing oil blend samples can be divided into three categories: (1) designing oil blend samples with equal intervals in a specified concentration range; (2) designing oil blend samples with unequal intervals in a specified concentration range; (3) designing oil blend samples according to the principle of design of experiments. It should also be noted that the order that the oil blend samples are presented to the analytical instruments should be randomized to prevent inadvertent linear correlations between the samples and instrument.

### 2.1. Equal Interval Percentage Design

If the oil blend samples are designed with equal intervals in a same concentration range for each oil, the total content of each oil in all samples should be equal. The procedure can be divided into two cases, according to whether the percentages of each oil are repeated in all samples or not. The first case is where the percentages are not repeated. If the minimum percentage of each oil is zero, the number of oil blend samples to be designated as *m*, and the number of oils is *p*, then the sum of the percentages for all samples is 100 *m*, and the sum of the percentages for each oil is 100 *m*/*p*. According to the arithmetic sequence summation formula, the percentage interval *d* of each oil can be calculated by *d* = 200/[(*m* − 1)*p*]. The maximum percentage of each oil can be calculated according to *d*, which is 200/*p*. It indicates that the percentage interval *d* is related to the number of samples *m* and number of oils *p*, while the maximum percentage is only related to the number of oils *p*. The maximum percentage of each oil for oil blends with a different number of component oils is provided in Table 1. It can be seen that the maximum percentage of each oil decreases with the increase in the number of component oils. In addition, the percentage of each oil in oil blends cannot reach 100%, with the exception of binary oil blends. In order to reduce the linear correlation between different component oils, the percentages of each oil is randomly distributed (using Matlab or similar software) to obtain the percentage design table for each oil blend sample under the constraint that the sum of oil percentages is 100.

The second case is where the percentages are repeated, and the percentage of each oil ranges from 0 to 100%. To design oil blend samples with equal intervals, the number of samples is related to the percentage interval *d* and the number of oils *p*. According to *d*, the number of gradients contained in each oil is *n* = 100/*d* + 1, then the number of samples *m_n_* for oil blends with a different number of component oils can be calculated. For a ternary oil blend,
(1)$$m_{n}=\sum_{i=1}^{n}C_{i}^{1}\;$$

Similarly, for a quaternary oil blend,
(2)$$m_{n}=\sum_{j=1}^{n}\sum_{i=1}^{n-j}C_{i}^{1}\;$$

The repeat times *P* of different percentages for the same oil is *P* = *m_n_* − *m_n_*_−1_. It should be noted that the decrease in the repeat times *P* corresponds to the increase in the percentages for each oil. For example, assuming that the percentage interval *d* is 10, then the number of gradients *n* is 11. Under this condition, the number of samples *m_n_* for ternary oil blends is 66. The percentages of each oil in 66 samples of ternary oil blends are listed in Table 2. The repeat times *P* for each oil of 0% is 11, for 10% is 10, and for 20% is nine. This method is used by most researchers. However, in this case, the sample size increases greatly with the decrease in the percentage interval *d* and the increase in the number of oils *p*. Under the same conditions, there would be 286 samples needed for quaternary oil blends, and 1001 samples needed for quinary oil blends.

### 2.2. Unequal Interval Percentage Design

Under the unequal interval percentage design, the contents of other oils are adjusted according to that of the targeted oil, thus the total content of each oil in all samples is not equal. Researchers usually use small percentage intervals at lower concentration, increasing to large percentage intervals at high concentrations. For instance, when preparing oil blend samples with *p* oils, the concentration of a certain oil may be designed to be 0%, 0.5%, 1%, 1.5%, 2%, 2.5%, 3%, 3.5%, 4%, 4.5%, 5%, 6%, 7%, 8%, 9%, 10%, 20%, 30%, 40%, 50%, 75%, 100%. The purpose of this method is to improve the sensitivity of the detection and the quantitative detection limit of the targeted oil. Nevertheless, as the percentage intervals are manually chosen, different intervals may provide different outcomes, thus this method is not universal.

### 2.3. Design of Experiments

The aforementioned methods are all designed to create oil blend samples with different percentages in a certain concentration range. However, according to the principle of design of experiments (DOE), only samples containing 0% and 100% of each pure oil are required. For example, as shown in Figure 2, a ternary oil blend was designed by simplex theory. The vertices of the equilateral triangle correspond to pure oils, and the edges correspond to all binary oil blends made up of the pure oils. Any point inside the simplex represents a ternary oil blend. For example, the point E represents a ternary oil blend containing 20% of oil 1, 30% of oil 2 and 50% of oil 3, which is determined by drawing lines pass through E and parallel to the edges. One of the greatest peculiarities of this method is that each point in the simplex satisfies the constraint that the sum of the oil contents is equal to 100% [10,11]. Given the complexity of edible oil compositions and the requirement of sample size for building models, it is necessary to conduct multiple tests on pure oil samples. Zhang et al. [12,13] attempted this method and found that the performance of the quantitative models established between the original pure oil spectral matrix and target values was not satisfactory. Consequently, the pure oil spectral matrix should be preprocessed by variable selection to improve prediction performance. To date, there have been relatively few studies applying DOE to design oil blend samples. However, in light of its universality and simplicity, this method is well worth the attention of future researchers.

The proportion of studies investigating edible oil blends with a different number of component oils in the literature from 2002 to 2022 is summarized in Figure 3. It can be seen that binary and ternary oil blends are the most studied so far, accounting for 72.7% and 19.5% of all work, while quaternary oil blends account for 6.5%. Almost no studies have focused on oil blends with a higher number of component oils, as the content of each oil is difficult to control with increasing oils.

The number of studies using different edible oil types for preparing edible oil blends is shown in Figure 4. It can be seen that there are over 20 studies on conventional edible oils such as soybean, sunflower and olive oils, while unconventional edible oils such as safflower, black seed and red fruit oils were only used once.

## 3. Instrumental Techniques

As shown in Figure 1, the prepared oil blend samples should be measured using different instruments. This section introduces several commonly used instrumental techniques for single component oil quantification in edible oil blends, including infrared, near-infrared, Raman, fluorescence, ultraviolet-visible, nuclear magnetic resonance spectroscopy and mass spectrometry. Chromatography, a common separation method, is usually used in combination with ultraviolet spectroscopy, diode array detector, nuclear magnetic resonance spectroscopy and mass spectrometry. Chromatography-based techniques use these detectors to determine the content of single component oil in edible oil blends by measuring the ratios of specific compounds, such as fatty acids or triacylglycerol [14,15,16]. Therefore, chromatography is not discussed in this paper.

### 3.1. Infrared Spectroscopy

Infrared (IR) spectroscopy, also referred to as mid-infrared (MIR) spectroscopy, commonly has a wavenumber range of 4000–400 cm^−1^. It provides an absorption spectrum of energy level transitions caused by molecular vibrations and rotations, which can provide information regarding molecular functional groups [2]. Fourier transform infrared (FTIR) spectrometer takes the incident light after Fourier transform, providing the advantages of high resolution, high sensitivity and fast scanning speed [9]. However, it is usually difficult to determine the molecular source of each peak due to the complex composition of edible oils [17]. Consequently, multivariate calibration methods combined with FTIR spectroscopy are used to analyze this complex information [18,19,20]. Rohman et al. [21,22,23] used FTIR spectroscopy coupled with PLS and PCR to establish models for the quantitative analysis of binary and ternary oil blends. The results showed that the models could achieve good prediction performance. Fadzlillah et al. [24] prepared 20 groups of binary oil blends of sesame and corn oils. They selected the 1072–935 cm^−1^ region of the FTIR spectra to build a PLS model to determine the corn oil content. Furthermore, de Souza et al. [25] combined FTIR spectroscopy with PLS models to quantify the contents of other oils in extra virgin flaxseed oil (EFO), so as to control the quality of EFO. The results showed that this method was suitable for quality control of EFO where the other oils were found in levels between 3.50% to 30% (*w*/*w*).

Oil blend samples analyzed by transmission FTIR spectroscopy need pretreatment. The samples should be diluted with solvents, which is a time-consuming process and produces additional solvent waste. However, it does not require any oil pretreatments by attenuated total reflectance FTIR (ATR-FTIR) spectroscopy. The spectra can be collected directly by placing the oil blend samples onto the ATR crystal without dilution [26]. ATR-FTIR spectroscopy obtains the structural information of the organic components through the IR signal reflected from the sample’s surface [27]. Jovic et al. [28] combined ATR-FTIR spectroscopy with PLS and PCR to quantify the content of single component oil in a ternary oil blend. The obtained limit of detection (LOD) for extra virgin olive oil (EVOO) was low to 0.93%. Akin et al. [29] combined ATR-FTIR spectroscopy with chemometrics to determine the content of soybean oil in grape seed oil. The content of soybean oil could be determined at a level <0.59% by PLS model. Both FTIR and ATR-FTIR techniques remain the commonly used instrumental techniques in the quantitative analysis of edible oil blends.

### 3.2. Near-Infrared Spectroscopy

Near-infrared (NIR) spectroscopy is a rapid, non-destructive and sensitive technology operating in the wavenumber range of 12,500–4000 cm^−1^. It has been widely applied for qualitative and quantitative analysis of agricultural products and foods [27]. NIR spectroscopy mainly records hydrogen-containing groups in organic molecules, such as O-H, N-H, C-H and S-H chemical bonds. However, the broad and often-overlapping bands caused by molecular overtones and combination vibrations make NIR spectra quite complex to interpret [2,30]. Therefore, chemometric methods must be used to extract useful chemical information.

The rapid development of chemometrics has promoted the development of NIR technology in quantitative analysis. To date, this technology has been widely used in the quantitative analysis of binary and ternary oil blends [31,32]. Feng et al. [33] selected the optimal NIR spectra region of 8745–4500 cm^−^^1^ to build calibration models to determine rapeseed oil content in sesame oil. The prediction results showed that the PLS models could achieve satisfactory results when the content of rapeseed oil was between 10% and 70%. Chen et al. [34] quantified the contents of four groups of binary oil blends by chemometric methods combined with NIR spectroscopy. The results indicated that this method could not only quantify different types of edible oils produced by the same manufacturer, but also quantify edible oils of the same type produced by different manufacturers. Liu et al. [35,36] determined the content of peanut oil in ternary and quaternary oil blends. They combined NIR spectroscopy with multivariate calibration methods to establish PLS, PCR and stepwise multiple linear regression (SMLR) models. The results showed that the prediction performance of the PLS model was superior.

### 3.3. Raman Spectroscopy

Raman spectroscopy is based on the Raman scattering effect. The Raman effect allows the acquisition of vibrational situations inside the molecule, thus allowing the characterization of different functional groups present. Furthermore, quantitative analysis models can be built based on the area or intensity of characteristic peaks [2,3]. Again, quantitative analysis is made more challenging due to spectral collinearity, and overlapping peaks, as well as the Raman intensity highly depends on the concentrations of the target analytes [37,38]. Therefore, it is necessary to combine advanced chemometric methods with Raman spectroscopy to improve the efficiency and accuracy of quantitative analysis.

Olive oil is usually more expensive than other oils due to its high contents of vitamins and antioxidants. Hence, many studies have focused on the quantitative analysis of olive oils [39,40,41]. de Lima et al. [42] used Raman spectroscopy and a mathematical method based on exponential equation fit to determine the volume fraction of rapeseed and corn oils found in olive oil samples. Li et al. [43] determined the proportion of waste cooking oil in olive oil using Raman spectroscopy. Interval partial least square (iPLS) and synergy interval partial least square (SiPLS) quantitative models were investigated. The results revealed the best theoretical LOD of approximately 0.48% for the SiPLS model. In addition, Dong et al. [44] used Raman spectroscopy combined with PLS to quantify the content of each oil in a quinary oil blend of peanut, sesame, rapeseed, soybean and corn oils. The above-cited studies demonstrate the broad applicability of Raman spectroscopy for the quantitative analysis of various oils in edible oil blends.

### 3.4. Fluorescence Spectroscopy

Fluorescence (FS) spectroscopy has the advantages of efficient, convenient and sensitive detection. ‘Fluorescence’ is a cold luminescence phenomenon of photoluminescence. It operates on the principle that after a substance absorbs electromagnetic radiation, the excited atoms or molecules return to their ground state. In this process of transitioning from a higher energy level to a lower energy level, energy is released in the form of electromagnetic radiation. The relationship between the FS energy and the corresponding wavelength is the FS spectrum. The content of a substance can be determined according to the FS intensity [45]. Due to the presence of various common fluorophores in edible oils, the FS spectra can be overlapped when analyzing oil blends [46]. Again, chemometric means are needed to extract and optimize the FS spectra to improve prediction performance [47,48].

Since it has a lower detection limit than other spectroscopic techniques [49], FS spectroscopy is a powerful tool to quantify single component oil in edible oil blends. Hu et al. [50] used synchronous fluorescence spectroscopy (SyFS) combined with PLS to quantify the contents of vegetable oils in Eucommia ulmoides seed oil. They selected the excitation spectral region between 300–500 nm to establish quantitative models. The LODs of the vegetable oils were as low as 0.48%. Poulli et al. [51] used total synchronous fluorescence (TSyF) spectroscopy to quantify the contents of olive-pomace, corn, sunflower, soybean, rapeseed and walnut oils in virgin olive oil. They used the excitation wavelengths between 250–720 nm and varied the wavelength interval in the region from 20–120 nm to obtain the TSyF spectra. The quantitative LODs of the six oils were 2.6%, 3.8%, 4.3%, 4.2%, 3.6% and 13.8% (*w*/*w*), respectively. Jing et al. [46] used FS spectroscopy to determine the each oil content in a ternary oil blend. An excitation-emission matrix (EEM) was collected to obtain comprehensive FS information in a short period of time, and the Quasi-Monte Carlo (QMC) integral was applied to quantify the concentrations of the three oils and their recovery rates. As with other studies using this analytical technique, the low LOD of FS spectroscopy was of great significance for the trace analysis of edible oil blends.

### 3.5. Ultraviolet-Visible Spectroscopy

Ultraviolet-visible (UV-vis) spectroscopy is a very common analytical technique that includes parts of the ultraviolet and visible light regions (200–800 nm). The resultant UV-vis spectrum is the result of electron energy level transitions in molecules or atoms, which absorb the UV-vis light. Different substances often show unique UV-vis spectra owing to their different compositions and spatial structures, although the peaks will often be overlapped due to the presence of common UV-active moieties. Particularly in complex matrices such as oil blends, it is difficult to directly use the spectra for quantitative analysis [52]. However, with the maturation of chemometrics as a discipline, many researchers have adopted UV-vis spectroscopy combined with chemometric techniques for single component oil quantification. In this way, the contents of specific substances can be quantified using the intensity of their characteristic UV-vis peaks [53].

UV-vis spectroscopy combined with chemometric methods has commonly been applied for quantitative analysis of EVOO. For example, Aroca-Santos et al. [54] used UV-vis spectroscopy coupled with chemometrics to quantify the volume percentage of EVOO in other oils. The established artificial neural network (ANN) model could not only discriminate the types of other oils added to EVOO, but could also achieve a satisfactory result in EVOO quantification. Additionally, these researchers also quantified the EVOO content of different brands [55] by constructing a MLR model and multilayer perceptron (MLP) model based on ANN. Another study by Jiang et al. [56] determined the content of EVOO in corn, soybean and sunflower oils using PLS models constructed from the UV spectra.

### 3.6. Nuclear Magnetic Resonance Spectroscopy

Nuclear magnetic resonance (NMR) is a physical process based on radio frequency radiation absorbed by atomic nuclei subjected to strong magnetic fields. Under a constant external magnetic field, the atomic nuclei with spin is irradiated by radio frequency radiation. When the radio frequency is exactly equal to the precession frequency of the atomic nuclei, it can be absorbed. The resulting resonance absorption spectrum is called a NMR spectrum [57]. NMR spectroscopy can directly provide the numbers of specific atoms in a sample present under different chemical environments, as well as the structural information of their adjacent groups. Consequently, it provides information about the molecular arrangement of organic samples [3].

With technical developments in automatic sampling, advancement in the resolution and speed of NMR spectrometers, and the development of new software and techniques for spectra processing, NMR spectroscopy has become an extremely powerful tool for analyzing edible oil blends. For instance, Jovic et al. [58] used ^1^H NMR spectroscopy to determine the concentration of adulterant oils in hempseed oil. The achieved low errors indicated that ^1^H NMR spectroscopy combined with chemometric methods could effectively quantify the adulterant levels. Alonso-Salces et al. [59] developed a stepwise strategy based on ^1^H-NMR fingerprinting to quantify the contents of other vegetable oils added to EVOO. This method achieved satisfactory results through blind sample testing. Smejkalova et al. [60] adopted high gradient diffusion NMR spectroscopy to measure the diffusion coefficients (D) of four oils in EVOO. The minimum adulteration levels of these oils could be determined by the changes of D. The results showed that the minimum adulteration levels were 10% for sunflower and soybean oils, and 30% for hazelnut and peanut oils. Although studies using NMR spectroscopy for single component oil quantification are not as numerous as those using other spectroscopic techniques, NMR spectroscopy is also worth the attention of future studies in this field.

### 3.7. Mass Spectrometry

Mass spectrometry (MS) is a technique used to identify unknown compounds in samples by preparing, separating and detecting gas-phase ions. It separates the gas-phase ions according to their mass-to-charge ratio. MS analyzes the structures of compounds by the position of their mass peaks, and can quantify compounds from the peaks intensities [2]. Zhou et al. [61] used thermogravimetric-gas chromatography/mass spectrometry (TGA-GC/MS) combined with chemometrics to determine the content of soybean oil in olive oil. Another study by Li et al. [62] used matrix-assisted laser desorption/ionization mass spectrometry (MALDI-MS) to analyze the triacylglycerol in oil blends, and built PLS models based on the MALDI-MS spectra to quantify the content of olive oil.

### 3.8. Other Methods

In recent years, the strategy of data fusion has gradually emerged in the field of spectral analysis. Uncu et al. [49] used data fusion to determine the content of old olive oil in fresh olive oil. The PLS model based on FT-IR + UV-vis data fusion achieved robust statistical parameters. Li et al. [63] investigated the combination of NIR and MIR spectroscopy for the quantification of rapeseed oil in olive oil. They used three data fusion strategies of low, mid and high-level to build PLS models. The results showed that the high-levels data fusion strategy could be used as a reliable tool for quantitative analysis. In addition, some less commonly used instrumental techniques have also been applied to quantify single component oil content. For example, Chen et al. [64] investigated the peak formation mechanism of vegetable oils using ion mobility spectrometry (IMS). They established a mobility spectral library of single component oil and edible oil blends, which was used to realize the single component oil quantification. Garrido-Delgado et al. [65] used a UV-IMS sensor in combination with multivariate calibration methods to determine the content of EVOO in other vegetable oils. Torrecilla et al. [66] combined lag-k autocorrelation coefficients (LCCs) with readings from a thermogravimetric analyzer (TGA) to quantify the contents of other vegetable oils in EVOO. Finally, Tsopelas et al. [67] used voltammetric fingerprinting of oil blends coupled with PLS for the quantitative analysis of olive pomace and seed oils in EVOO.

The advantages and disadvantages of the various analytical techniques are summarized in Table 3. This should help researchers to choose appropriate instrumental techniques for the quantitative analysis of single component oil. As can be seen in Figure 5, spectroscopic techniques remain the main instruments for the quantitative analysis of oil blends, particularly IR and NIR spectroscopy. The rapid development of chemometrics promotes the use of IR and NIR spectroscopy in the quantitative analysis of complex systems. Besides these techniques, FS spectroscopy can be a powerful tool in single component oil quantification due to its low detection limit. In recent studies, prediction results obtained by a data fusion strategy of different spectra have been shown to yield better results than that of individual spectra, hence this strategy provides a new idea for single component oil quantification in edible oil blends.

## 4. Chemometric Methods

The final step in the analysis of oil blend samples is processing of the obtained datasets using chemometric methods for signal pretreatment and quantitative model establishment. Chemometrics is used to extract pertinent information from spectra and reduce background interference. It mainly includes preprocessing, variable selection and multivariate calibration. This section introduces some popular chemometric methods used for single component oil quantification in edible oil blends.

### 4.1. Preprocessing Methods

In addition to possessing useful chemical information of samples, the measured matrix also contains some useless information and noise, which can affect the accuracy of quantitative analysis. Therefore, signal pretreatment is necessary to eliminate the influence of useless information and noise before constructing quantitative models. The frequency of use for different preprocessing methods is shown in Figure 6a. These include mean centering (MC), normalization, smoothing, derivative, standard normal variate (SNV) transformation and multiplicative scatter correction (MSC).

MC subtracts the average spectrum of the calibration set from the sample spectrum and removes the common information found in all spectra. This method can improve the stability of models [68,69]. Spectral normalization is usually used to eliminate effects caused by the changes in light path and sample dilution [47,70]. The purpose of smoothing is to reduce the noise from spectra signal. Savitzky–Golay (SG) smoothing is widely used to improve the appearance of peaks that are obscured by noise [69,71]. Taking the derivative is the most commonly applied preprocessing method in quantitative analysis. It can be used for baseline correction by deducting the influence of the instrument background or signal drift [72,73]. SNV is used to reduce the multiplicative effect of different solid particle sizes and surface scattering on NIR or IR spectra. The spectra transposed by this method are free from multi-collinearity [74]. The purpose of MSC is similar to that of SNV, in that it can reduce the effect of spectra caused by solid particle size and uneven particle distribution. Under certain conditions, SNV and MSC are interconvertible [75,76].

In the quantitative analysis of single component oil in edible oil blends, the measured matrix is usually preprocessed before model establishment to increase the accuracy and reliability of the results [77,78,79,80]. Du et al. [81] used MSC, SNV, SG smoothing, Norris derivative and normalization pretreatments in the quantification of other oils in camellia oil. Ding et al. [82] performed pretreatments of offset, offset + SNV, offset + second derivative and offset + SNV + second derivative on IR spectra to quantify the concentrations of soybean and sunflower oils in ternary oil blends. Li et al. [83] applied continuous wavelet transform, smoothing and first derivative pretreatments on NIR spectra before building PLS models to perform quantitative analysis of a quaternary oil blend. Compared with no preprocessing, the prediction performance of the models was improved through optimal preprocessing methods. Furthermore, Bian et al. [84] proposed a selective ensemble preprocessing strategy. The results demonstrated that this strategy could achieve comparable or even better results than the best preprocessing method selected through traditional means.

### 4.2. Variable Selection Methods

The measured matrix usually contains hundreds of variables due to the complex composition of edible oil blends. Each of these variables may be informative, uninformative or just represent inter-correlated variables. Additionally, using a large number of variables and small number of samples may cause overfitting problems [85,86]. Thus, it is necessary to select the most informative variables before constructing quantitative models. The process of variable selection aims to choose a small number of variables, which relate to the properties of interest to improve the prediction performance of the models. Previous reviews by Mehmood et al. [87] and Yun et al. [88] introduced and classified spectral variable selection methods from different perspectives.

Competitive adaptive reweighted sampling (CARS) and bootstrapping soft shrinkage (BOSS) are the two most commonly used variable selection methods. CARS selects *N* subsets of wavelengths from N Monte Carlo (MC) sampling runs in an iterative and competitive manner. In each sampling run, a fixed proportion (e.g., 80%) of samples is first randomly selected to build a calibration model. Then, the optimal combination of wavelengths is selected through an exponentially decreasing function (EDF) and adaptive reweighted sampling (ARS) [89]. BOSS, a new variable selection method proposed in recent years, is used to select informative variables where collinearity exists. It is developed from the idea of weighted bootstrap sampling (WBS) and model population analysis (MPA). WBS is used to generate sub-models according to the weights, and MPA is used to analyze the sub-models to update weights for variables. This algorithm follows the rule of soft shrinkage, wherein the less important variables are not directly eliminated but are assigned smaller weights. This method runs in an iterative manner until the number of variables reaches one [90].

Although variable selection methods are not as commonly used as preprocessing methods, it is undeniable that the prediction results of quantitative models are more accurate and reliable after variable selection. Basri et al. [80,91] used NIR spectroscopy to determine the lard content in palm oil. In order to remove the uninformative variables, the CARS method was applied. The results after variable selection for both transflectance and transmission spectra were improved significantly. Chen et al. [92] performed CARS to select 10 variables from NIR spectra for use in PLS regression. They successfully quantified the contents of other oils in sesame oil. Jiang et al. [30] applied the BOSS algorithm to select the optimal variable subset for PLS modeling to determine the contents of other oils in EVOO, which include 15 wavenumbers. Compared with the optimal models of CARS-PLS, Monte Carlo uninformative variable elimination PLS (MCUVE-PLS), and iteratively retaining informative variables PLS (IRIV-PLS), the predictive ability of BOSS-PLS was the best. In addition, Ruiz–Samblas et al. [93] adopted PLS to build regression models relating the triacylglycerol profiles of oil blends to quantify the content of olive oil. Genetic algorithm (GA) was used as a variable selection method to improve the model, with the GA-PLS model showing improved predictive ability. With the development of variable selection methods, different combinations or hybridizations of different algorithms are attracting increasing attention from specialists and scholars [72].

### 4.3. Multivariate Calibration Methods

Multivariate calibration methods, including linear and nonlinear calibration methods, are often used for model development in the quantitative analysis of single component oil in edible oil blends. The premise of linear calibration methods is that the spectral matrix has linear additivity, that is, it obeys the Lambert–Beer law. It mainly includes MLR, PCR, and PLS models. However, in practice, there is not always a linear relationship between the spectral matrix and the target values due to instrument noise, baseline drift and other issues. In this case, nonlinear calibration models need to be established. The most commonly used nonlinear calibration methods include ANN, SVR and extreme learning machine (ELM).

Different statistical criteria are used to evaluate the performance of calibration models. These include the coefficient of determination (R^2^), root mean square error of calibration (RMSEC) for the calibration set, root mean square error of prediction (RMSEP) for the prediction set, and root mean square error of cross-validation (RMSECV) for the cross-validation set. R^2^ represents the percentage of response variables, which can be explained by spectral matrix; the closer R^2^ is to one, the closer the predicted value is to the actual value. RMSEC is used to evaluate the feasibility of modeling, RMSEP is used to evaluate the predictive ability of the established model to external samples, and RMSECV is used to evaluate the degree of fitting during the cross-validation process. An ideal model should have a high R^2^ value, low RMSEC, RMSEP and RMSECV values. In addition, the residual predictive deviation (RPD) can also be used to evaluate the performance of models. The RPD value of a good model should be at least 2.5 [94,95].

#### 4.3.1. Linear Calibration Methods

Multiple regression analysis refers to the use of regression equations to quantitatively explain the linear relationship between the dependent variable and two or more independent variables. MLR is a linear calibration method, which is easy to calculate and generally shows good statistical properties. It is suitable for systems with simple linear relationships, and where there are no mutual effects between components. However, MLR can only be applied to situations where the number of variables is less than the number of samples. Furthermore, when the degree of correlation between independent variables in dataset is too high, multi-collinearity makes it difficult to achieve satisfactory prediction results [96,97]. Finally, it often leads to overfitting due to not considering the noise, which exists in the spectral matrix **X**.

In many spectral matrices, the number of variables is usually more than the number of samples. In these situations, MLR is not feasible. PCR, which combines principal component analysis (PCA) with MLR to overcome collinearity, is more suited to these complex systems [98]. It calculates the final prediction equation by diagnosing collinearity between independent variables [96,99]. Jamwal et al. [100] combined IR spectroscopy with PCR to determine the linseed oil content in mustard oil. The predictive R^2^ values of the established PCR models ranged from 0.979 to 0.998, while the RMSEP ranged from 1.51% to 0.46% *v*/*v*, and the highest RPD was 23.02. Wang et al. [101] analyzed the fatty acid compositions of peony seed oil based on its characteristic Raman peaks. The established PCR model successfully quantified the content of peony seed oil in other oils with a small residual error.

PLS was first proposed by S. Wold in 1982 and used in chemical applications [102]. It brings together the concepts of MLR, PCR and canonical correlation analysis. PLS not only has the advantages of PCR, but also considers the concentration information of samples that is not considered by PCR. It combines the decomposition and regression of the spectral matrix **X** and concentration matrix **Y** (or vector **y**) to increase the model accuracy and stability [103,104]. In spectral analysis, PLS can complete the orthogonal decomposition of the spectral matrix **X** and the concentration matrix **Y** (or vector **y**), and simulate a linear relationship between the spectral matrix **X** and the chemical composition [105,106]. Generally, PLS has low computational errors, good predictive ability and high accuracy. In the calibration problems, PLS outperforms MLR, PCR and ridge regression (RR).

For the quantitative analysis of single component oil in edible oil blends, PLS and improved PLS are the most widely used calibration methods. Numerous previous studies [107,108,109,110,111,112,113,114] have applied spectroscopic techniques combined with PLS to quantify the contents of component oils. The R^2^ values of the models were all greater than 0.95. One study demonstrated that the improved model of N-way partial least square (N-PLS) [115] could achieve good results with R^2^ > 0.98, and RMSECV < 3.91%. Jović investigated binary oil blends, finding that the established first-break forward interval PLS (FB-FiPLS) model could successfully quantify the contents of other oils in hempseed oil with R^2^ > 0.995, RMSECV and RMSEP in the range of 0.9%–2.9% and 0%–3.2% [116]. The established Durbin–Watson PLS (dwPLS) model could successfully quantify the contents of other oils in cold-pressed linseed oil. The RMSECV ranged from 1.2% to 2.6%, the RMSEP ranged from 1.3% to 2.5%, and the LOD ranged from 0.85% to 1.69% [117].

#### 4.3.2. Nonlinear Calibration Methods

ANN is a nonlinear calibration method developed by attempting to simulate the human brain. The connections between the neurons form a network, which can store and calculate information. It has the advantages of self-learning, self-adaptation and self-organization. The purpose of ANN training is to minimize the prediction error of neural network through altering the weights and biases of different connections [118]. The most widely used neural network is back propagation-artificial neural network (BP-ANN). It is a multilayer feed-forward network and comprises an input layer, hidden layer(s) and an output layer. Usually, the transfer functions used by BP-ANN neurons are sigmoid differentiable functions or linear functions [119,120]. This method has excellent nonlinear mapping approximation ability and prediction performance. Aroca-Santos et al. [121] coupled UV-vis spectroscopy with ANN for the quantitative analysis of binary oil blends. For oil blends with refined olive oil content between 0% and 20%, the quantitative mean prediction error for EVOO was 2.14%.

Support vector machine (SVM) is a pattern recognition method proposed by Vapnik [122] in 1995. It uses an optimal hyperplane to separate two sample classes without error [123]. The main idea of SVM is to utilize a kernel function to map low-dimensional input variables into a high-dimensional feature space through nonlinear transformation, and then to perform a linear solution in the feature space to generate a linear regression equation [105,106]. At present, three types of kernel functions are commonly used, polynomial, radial basis, and S-shaped kernel functions [124]. In recent years, SVM has been popularized for nonlinear regression and function approximation in spectral analysis. In some situations, the predictive accuracy of SVR is higher than that of PLS [125]. Zhang et al. [126,127] developed two new algorithms of particle swarm optimization least square support vector machine (PSO-LSSVM) and quantum-behaved particle swarm optimization multi-output least square support vector machine (QPSO-MLSSVM) to realize the quantitative analysis of single component oil in ternary and quaternary oil blends. The predictive mean square errors of these two algorithms were both less than 0.1%.

ELM is a single-hidden layer feed-forward neural network, which has characteristics of easy parameter selection, fast learning speed and good generalization ability. In nonlinear methods, ELM is much faster than BP-ANN, k-nearest neighbor (K-NN), and least square SVM (LS-SVM) [128]. ELM randomly generates the connection weights between the input layer and hidden layer. In addition, the hidden layer does not need to be adjusted during the training process [129]. It only needs to optimize the number of hidden layer nodes and the activation function to obtain the unique optimal solution. However, ELM suffers from reduced stability and robustness due to the random generation of input weights and hidden layer biases [130]. A boosting ELM proposed by Bian et al. [131] was used for quantitative analysis of edible oil blends. This method established a large number of ELM sub-models according to the distribution of the sampling weights, and generated the final result by aggregating the prediction results of these sub-models through the weighted median. The accuracy and stability of the boosting ELM were both higher than those of ELM and PLS.

The advantages and disadvantages of multivariate calibration methods are summarized in Table 4, to aid researchers in comparing these methods. Additionally, Figure 6b shows the frequency of occurrence of different calibration methods in the literature. It can be seen that PLS is by far the most preferred calibration method for quantitative analysis of single component oil in edible oil blends, being used in around 75.3% of relevant studies. Although it is the least used method to date, ELM appears worthy of further exploration in the single component oil quantification, due to its fast learning speed and broad generalization ability. The concept of an ensemble strategy has also come to the fore in recent years. Ensemble modeling combines the predictions from multiple sub-models to obtain a more accurate, stable and robust prediction. Hence, new and effective ensemble modeling methods should be developed.

## 5. Conclusions and Perspectives

In order to determine the content of single component oil in edible oil blends, various analytical methods combined with chemometric techniques have gained attention from researchers. These analytical techniques require a large number of oil blend samples with a different number of component oils and proportions to establish quantitative models. This review first summarized the three sample design methods used in studies: (1) designing oil blend samples with equal intervals in a specified concentration range; (2) designing oil blend samples with unequal intervals in a specified concentration range; (3) designing oil blend samples following the principles of DOE. Among these, the use of DOE to design edible oil blend samples is worth trying in future studies, due to its universality and simplicity.

Spectroscopic technology has become the main instrumental technique in the quantitative analysis of single component oil in edible oil blends. Among them, IR and NIR spectroscopy are still the most widely used instruments. In addition to these, FS spectroscopy and the use of data fusion are worth future exploration due to their low detection limit and excellent predictive ability, respectively. Chemometrics is an indispensable step in the process of single component oil quantification due to its ability to extract key spectral information and reduce background interference. Among the multivariate calibration methods, PLS remains the most widely used method. However, ELM shows promise as an alternative prediction method due to its fast learning speed and good generalization ability. In addition, preprocessing methods such as derivative, SNV and smoothing are used to increase model accuracy and robustness, while CARS and BOSS are the commonly used variable selection methods. With the maturation of chemometric techniques, applying the idea of ensemble strategy into preprocessing, variable selection and multivariate calibration can help researchers obtain more stable, accurate and robust prediction results.

At present, most researchers investigate single component oil quantification in edible oil blend samples with certain component oils and concentrations, while few researchers directly determine the content of single component oil in edible oil blends purchased from supermarkets. In practice, it is feasible for edible oil industries to use these analytical techniques to determine the edible oil blends produced by themselves (i.e., for quality assurance purposes). However, the component oils and concentrations of the most edible oil blends are unknown in supermarkets, which means these techniques have limitations in determining the completely unknown edible oil blends in practice. Therefore, there is still much groundwork to be done before these methods can be used in practical applications. In addition, it is always necessary to establish a quantitative model corresponding to the target component oil whether in binary or ternary oil blends, which is a cumbersome and time-consuming process. If a general model can be developed for the quantitative analysis of a certain conventional basic edible oil in edible oil blends with different number of component oils, it would provide a great benefit to future workers.

## Figures and Tables

**Figure 1 foods-11-02436-f001:**
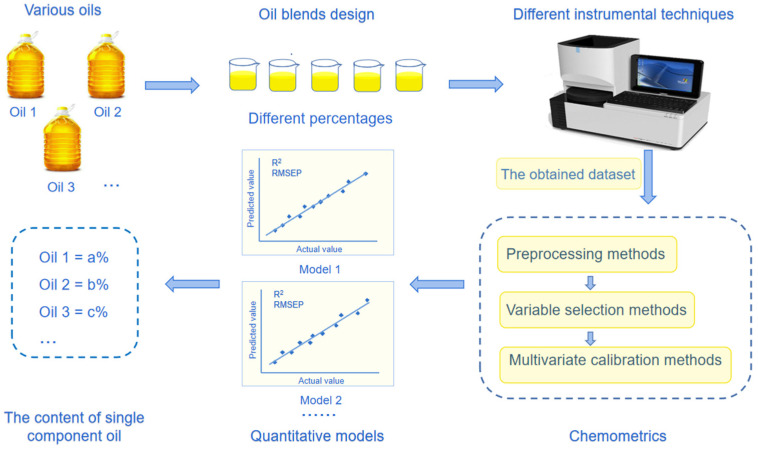
The general process for single component oil quantification in edible oil blends.

**Figure 2 foods-11-02436-f002:**
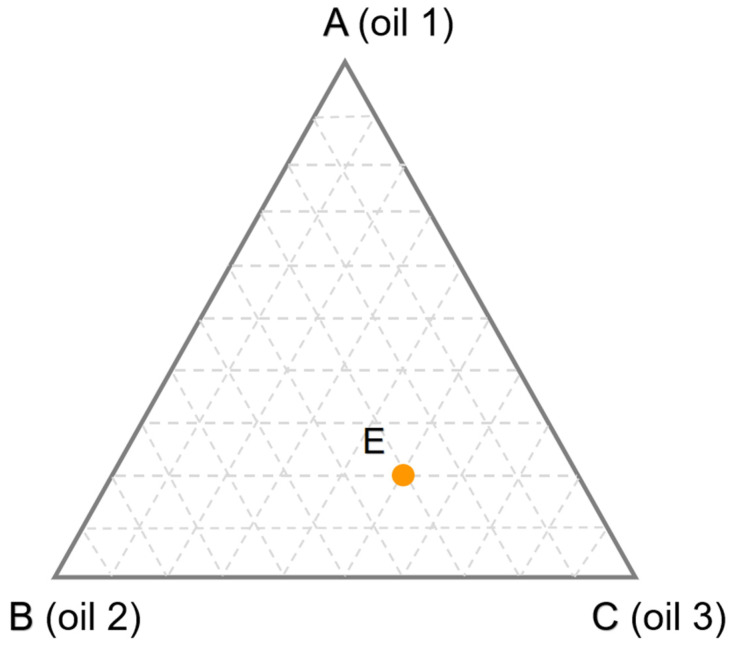
The theory of simplex for designing a ternary oil blend. The point E represents a ternary oil blend containing 20% of oil 1, 30% of oil 2 and 50% of oil 3, which is determined by drawing lines pass through E and parallel to the edges.

**Figure 3 foods-11-02436-f003:**
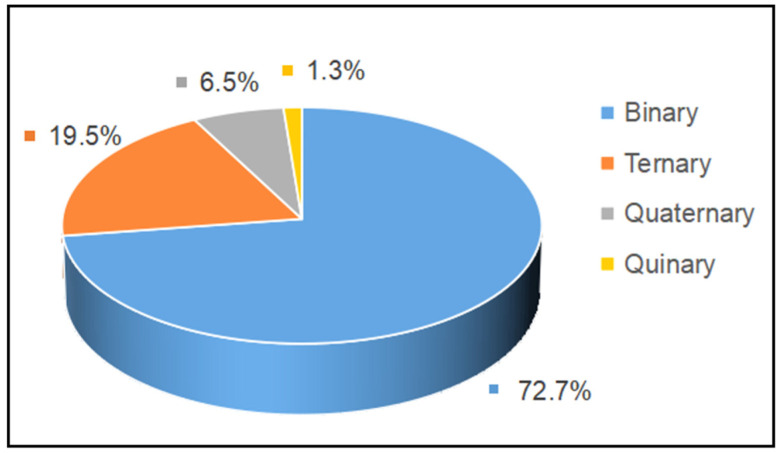
The proportion of studies investigating edible oil blends with different number of component oils in literature from 2002–2022.

**Figure 4 foods-11-02436-f004:**
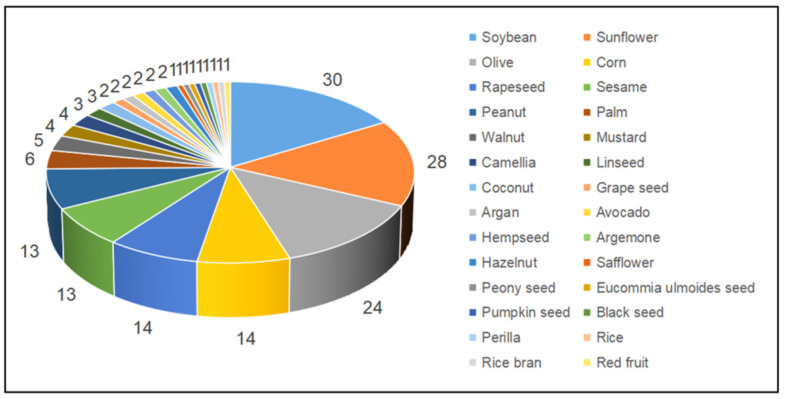
The number of studies using different edible oils for preparing edible oil blends in literature from 2002–2022.

**Figure 5 foods-11-02436-f005:**
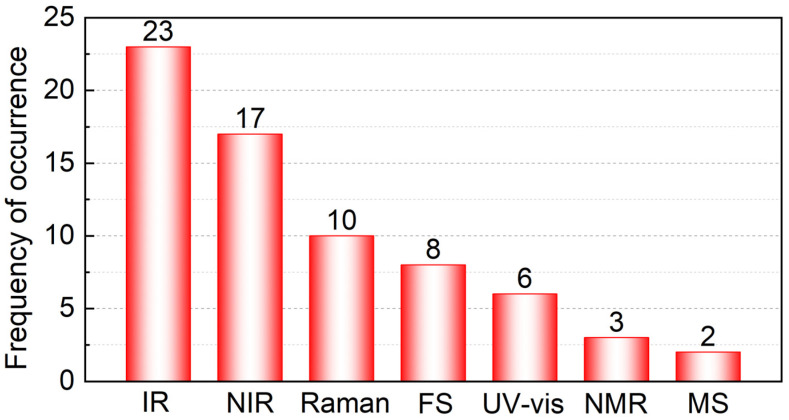
Frequency of occurrence in the literature for different analytical techniques.

**Figure 6 foods-11-02436-f006:**
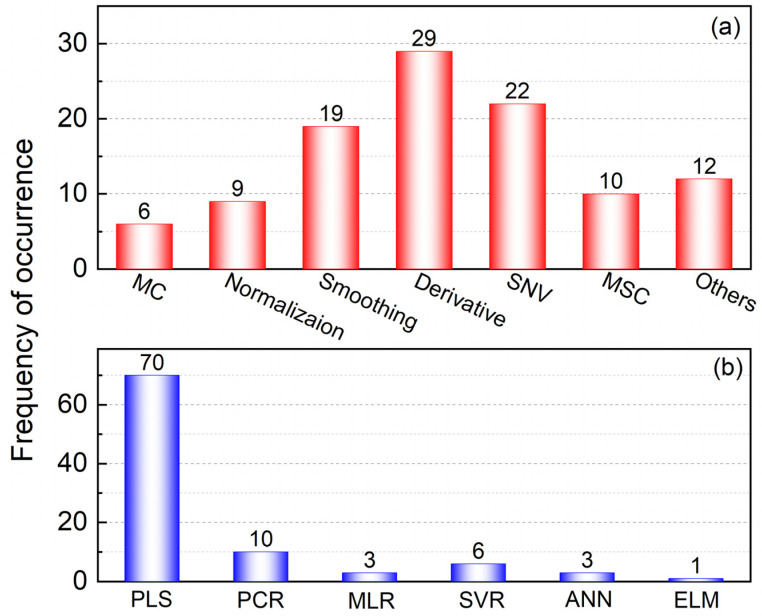
Occurrence of studies using preprocessing: (**a**) and calibration; (**b**) methods in relevant literature.

**Table 1 foods-11-02436-t001:** The maximum percentage of each oil for edible oil blends with different number of component oils.

The Number of Component Oils	Minimum Percentage	Maximum Percentage
2	0	100
3	0	66.7
4	0	50
5	0	40
6	0	33.3
7	0	28.6
8	0	25
9	0	22.2
10	0	20

**Table 2 foods-11-02436-t002:** The percentages of each oil in ternary oil blends.

No.	Oil 1	Oil 2	Oil 3	No.	Oil 1	Oil 2	Oil 3	No.	Oil 1	Oil 2	Oil 3
1	0	0	100	23	20	10	70	45	40	60	0
2	0	10	90	24	20	20	60	46	50	0	50
3	0	20	80	25	20	30	50	47	50	10	40
4	0	30	70	26	20	40	40	48	50	20	30
5	0	40	60	27	20	50	30	49	50	30	20
6	0	50	50	28	20	60	20	50	50	40	10
7	0	60	40	29	20	70	10	51	50	50	0
8	0	70	30	30	20	80	0	52	60	0	40
9	0	80	20	31	30	0	70	53	60	10	30
10	0	90	10	32	30	10	60	54	60	20	20
11	0	100	0	33	30	20	50	55	60	30	10
12	10	0	90	34	30	30	40	56	60	40	0
13	10	10	80	35	30	40	30	57	70	0	30
14	10	20	70	36	30	50	20	58	70	10	20
15	10	30	60	37	30	60	10	59	70	20	10
16	10	40	50	38	30	70	0	60	70	30	0
17	10	50	40	39	40	0	60	61	80	0	20
18	10	60	30	40	40	10	50	62	80	10	10
19	10	70	20	41	40	20	40	63	80	20	0
20	10	80	10	42	40	30	30	64	90	0	10
21	10	90	0	43	40	40	20	65	90	10	0
22	20	0	80	44	40	50	10	66	100	0	0

**Table 3 foods-11-02436-t003:** The advantages and disadvantages of different analytical techniques used in the quantitative analysis of edible oil blends.

Techniques	Advantages	Disadvantages
IR spectroscopy	Simple and fast Strong absorbance High sensitivity No solvent for ATR-IR	Need solvent for traditional IR
NIR spectroscopy	Simple and fast Less or no solvent	Overlapping bands, background and weak absorbance
Raman spectroscopy	High efficiency Preprocessing-free for sample	Peaks overlapping Low sensitive for fluorescent and colored sample
FS spectroscopy	Low detection limit Fast and accurate High selectivity	Peaks overlapping Many measurement parameters
UV-vis spectroscopy	Fast and cheap No solvent Real-time analysis	Big noise in 200–400 nm Only a few peaks
NMR	Fast and effective High selectivity and accuracy	Need solvent
MS	Sensitivity Simple and accurate	Time-consuming Sample destructive

**Table 4 foods-11-02436-t004:** The advantages and disadvantages of multivariate calibration methods.

Methods	Advantages	Disadvantages
MLR	Simple calculation No parameter	Number of samples should be more than that of variables
PCR	Good predictive ability Only one parameter	Concentration information is not considered in the dimensionality reduction process
PLS	Fast calculated speed Only one parameter Good predictive ability High accuracy	Unsuitable for nonlinear problems
ANN	Self-learning Self-adaptation Excellent nonlinear mapping approximation ability	Difficult convergence Unstable solution Poor generalization ability Over-fitting
SVR	Suitable for pattern recognition and nonlinear high dimensional space problems	Small number of samples Three parameters need to be optimized
ELM	Fast learning speed Good generalization ability	Low stability and robustness

## Data Availability

Data is available from corresponding author upon request.

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
