# Peer review of "A Review of Advanced Methods for the Quantitative Analysis of Single Component Oil in Edible Oil Blends"

_foods, 2022, doi:10.3390/foods11162436_

Round 1
Reviewer 1 Report
1. The work is interesting and provides some novel findings regarding the existing methods for quantitative analysis of edible blend oils
2. Please describe what makes this article unique among other articles written on the same topic, and why there is still a need for a new article on the topic
3. It is necessary to improve the quality of figure 3 for a better presentation
4. The introduction is good, but it could be improved.
5. It would be helpful if you could explain how edible oil industries determine the most appropriate sample design methods in order to obtain blend oil samples.
Reviewer 2 Report
The review by Bian et al. reports the classification and comparison of the existing methods for quantitative analysis of edible blend oil. However, Extensive editing of the English language and style is required.
Moreover, other specific comments are listed below:
- Introduction: Line 72- 79, please modify the paragraph stating more clearly the aim of the review.
- I do not think section " 3.7 Chromatography" is needed. Probably the chromatogram can be described, in the introduction of section 3, as a common separation method applied to almost all the spectroscopic and spectrometric methods described in the other sections. HPLC and GC are usually used hyphenated with UV, DAD, Mass Spec, and NMR, but the detection or quantification of the components is obtained according to the response of the detectors.
- In Section "5. Conclusion and perspectives", could be
worthwhile to discuss more deeply the advantages and disadvantages of each analytical technique, as well as those of each Chemometric method used. Probably also organising a summary table could be
worthwhile for the readers.
Reviewer 3 Report
Comments to authors
Journal: Foods-1815057
Title: ‘Advanced methods review for quantitative analysis of single component oil in edible blend oil’
This current review article elaborates the detail of individual oil in the edible blend oil, the designed methods, instrumental techniques and chemometric methods for the quantification of single component of oil. This review has certain scientific significance.
I go through the article I recommend to resubmit this article as it needs deep revision from English experts as I mentioned fewer comments below, by improving the concerned point the article could be suitable for publication.
Minor comments:
1. Line 16 ‘This review summarizes… the sentence should be revised.
2. Line 24 ‘Edible oil is an important part in human daily life, which is widely used in cooking and flavoring’ sentence should be revised.
3. Line 31 ‘Besides, many researches have proved…. Rewrite the sentence.
4. Line 33 ‘Hence, compared with single edible oil, the sales proportion of edible blend oil continues to rise’ revise the sentence.
5. Line 44 ‘deception’ revise this word.
6. Line 58 ‘The obtained spectra usually contain…. Authors should improve the sentence by the linkage with the previous sentence, or remove it.
7. The paragraph from line 61-71 need to be revised.
8. Line 75 ‘Finally, the popular preprocessing, variable selection and multivariate calibration methods used for measured signal pretreatment…. This sentence has no meaning kindly revise it.
9. Line 96 ‘When preparing different order blend oil samples, the design of each oil contents a difficulty’ revise this sentence.
Main comments:
1. I strongly recommend the author to revised the article thoroughly a native English speaker. It has plenty of syntax as well as grammatical errors.
2. The overall article need deep revision with well connection among the sentences and flow of the article is difficult to understand.
3. Introduction of the manuscript should be improved and well-structured there are many grammatical errors throughout the manuscript.
4. Although there are many review articles are been published for the analysis of single oil from blended oil, so what is the innovation, purpose and explanatory techniques in these reviews?
5. Provide the reference against this sentence line 84 ‘Finally, the popular preprocessing, variable selection and multivariate calibration methods used for measured signal pretreatment and model establishment are also summarized’.
Round 2
Reviewer 2 Report
The authors revised the manuscript according to reviewers comments and suggestions and the manuscript can be accepted in the present form.
Author Response
Thank you very much for your valuable comments on our manuscript. We benefit greatly from you. We have responded the comments carefully one by one. Please find it in attachment.

Reviewer 3 Report
Comments to authors
Journal: Foods-1815057
Title: A review of advanced methods review for the quantitative analysis of single component oil in edible oil blends oil
This current revised review article has been greatly improved; still, I have fewer concerns as below.
1. Why the authors did not include cost/production analysis by comparing these methods would be better for the readers to understand which techniques is best for them.
2. During blending of oils, the content of each oil is equal why?
3. Does it required any pretreatments for oils before applying Infrared spectroscopy methods?
Author Response
Thank you very much for your valuable comments and suggestions on our manuscript. The manuscript has been revised according to your advice. We have responded the comments carefully one by one. Please find it in attachment.
